# Status of Insecticide Resistance and Its Mechanisms in *Anopheles gambiae* and *Anopheles coluzzii* Populations from Forest Settings in South Cameroon

**DOI:** 10.3390/genes10100741

**Published:** 2019-09-24

**Authors:** Roland Bamou, Nadège Sonhafouo-Chiana, Konstantinos Mavridis, Timoléon Tchuinkam, Charles S. Wondji, John Vontas, Christophe Antonio-Nkondjio

**Affiliations:** 1Vector Borne Diseases Laboratory of the Applied Biology and Ecology Research Unit (VBID-URBEA), Department of Animal Biology, Faculty of Science of the University of Dschang, Dschang P.O. Box 067, Cameroon; timotchuinkam@yahoo.fr; 2Laboratoire de Recherche sur le Paludisme, Organisation de Coordination pour la lutte contre les Endémies en Afrique Centrale (OCEAC), Yaoundé B. P.288, Cameroun; nadou_chiana33@yahoo.fr; 3Faculty of Science, University of Buea, Buea P.O. Box 63, Cameroon; 4Institute of Molecular Biology and Biotechnology, Foundation for Research and Technology-Hellas, 70013 Heraklion, Greece; mavridiskos@gmail.com (K.M.); vontas@imbb.forth.gr (J.V.); 5Vector Biology Liverpool School of Tropical medicine Pembroke Place, Liverpool L3 5QA, UK; charles.wondji@lstmed.ac.uk; 6Centre for Research in Infectious Disease (CRID), Yaoundé P.O. Box 13591, Cameroun; 7Pesticide Science Laboratory, Department of Crop Science, Agricultural University of Athens, 11855 Athens, Greece

**Keywords:** insecticide resistance, G119S mutation, malaria, anopheles, South Cameroon forest region

## Abstract

A key factor affecting malaria vector control efforts in Cameroon is the rapid expansion of insecticide resistance in *Anopheles gambiae* s.l *(An. gambiae*) populations; however, mechanisms involved in insecticide resistance in forest mosquito populations are still not well documented yet. The present study was conducted to screen molecular mechanisms conferring insecticide resistance in *An. gambiae* s.l. populations from the South Cameroon forest region. WHO bioassays were conducted with F0 *An. gambiae* females aged three to four days from forest (Sangmelima, Nyabessan, and Mbandjock) and urban sites (Yaoundé (Bastos and Nkolondom)), against pyrethroids (permethrin 0.75% and deltamethrin 0.05%) and carbamates (bendiocarb 0.1%). Members of the *An. Gambiae* s.l. species complex were identified using molecular diagnostic tools. TaqMan assays were used to screen for target site mutations. The expression profiles of eight genes implicated in insecticide resistance were assessed using RT-qPCR. Cuticle hydrocarbon lipids were measured to assess their potential implication in insecticide resistance. Both *An. Gambiae* and *An. coluzzii* were detected. *An. gambiae* was highly prevalent in Sangmelima, Nyabessan, Mbandjock, and Nkolondom. *An. coluzzii* was the only species found in the Yaoundé city center (Bastos). Low mortality rate to both pyrethroids and bendiocarb was recorded in all sites. High frequency of L1014F allele (75.32–95.82%) and low frequencies of L1014S (1.71–23.05%) and N1575Y (5.28–12.87%) were recorded. The G119S mutation (14.22–35.5%) was detected for the first time in *An. gambiae* populations from Cameroon. This mutation was rather absent from *An. coluzzii* populations. The detoxification genes *Cyp6m2*, *Cyp9k1*, *Cyp6p4*, *Cyp6z1*, as well as *Cyp4g16* which catalyzes epicuticular hydrocarbon biosynthesis, were found to be overexpressed in at least one population. The total cuticular hydrocarvbon content, a proxy of cuticular resistance, did not show a pattern associated with pyrethroid resistance in these populations. The rapid emergence of multiple resistance mechanisms in *An. Gambiae* s.l. population from the South Cameroon forest region is of big concern and could deeply affect the sustainability of insecticide-based interventions strategies in this region.

## 1. Introduction

According to the 2018 World malaria report, Cameroon is classified as one of 11 countries most affected by malaria in the world [1]. The whole country is exposed to malaria transmission risk with high and perennial transmission occurring in the forest region [2,3]. *An. gambiae* which was absent from forested zones has now invaded the area [4,5]. The rapid deforestation taking place in the forest region has provided suitable breeding opportunities for *An. gambiae*, which in most forest settings is now becoming the predominating species responsible for most malaria transmission cases [6,7,8,9]. In Cameroon, members of the *An. gambiae* complex consist of *An. gambiae*, *An. coluzzii*, *An. melas*, and *An. arabiensis*, but only *An. gambiae* and *An. coluzzii* have been reported from forest settings [4,5]. Although local forest anopheline species are still highly susceptible to insecticides due to their high exophilic and exophagic behavior [10], studies conducted so far suggested increased insecticide resistance in *An. gambiae* s.l. populations [8,11]. However, there is still not enough information on mechanisms conferring resistance to these forest *An. gambiae* s.l. populations. 

The rnapid spreading of insecticide resistance in main malaria vectors, particularly *An. Gambiae* s.l., is considered to impair the effectiveness of control efforts [12,13]. A recent review conducted in Cameroon indicated high prevalence of pyrethroid and DDT resistance almost everywhere [11]. Carbamate resistance is also present but at a limited scale [11,14]. According to the World Health Organization, over 15 compounds are considered to have been regularly used for vector control across the world. This includes DDT (organochlorine), bendiocarb and propoxur (carbamates), fenitrothion, malathion, and temephos (organophosphate), alpha-cypermethrin, bifenthrin, cyfluthrin, cypermethrin, cyphenothrin, deltamethrin, etofenprox, lambda-cyhalothrin, and permethrin (pyrethroid) [15]. The spread of pyrethroid resistance is thought to have resulted from the intensification of vector control measures on the field and pesticide use in agriculture [16,17,18]. In Cameroon, pyrethroid treated nets are the main tools used for malaria prevention. During the last decade two important mass distribution campaigns of LLINs have been conducted respectively in 2011 and 2015 and permitted the distribution of over 20 million nets to the population [2]. Concerning agriculture, with the increasing demands for new or exotic products, (fruits or vegetables), or the introduction of new agricultural practices, there has been increased usage of pesticides in agriculture in most forest settings [16,19]. How these practices are now affecting the bionomic of vector populations is not well documented. Two main mechanisms confer resistance to insecticides: Target site resistance and metabolic resistance. Target site resistance involving both L1014F (kdr West Africa) and L1014S (kdr East Africa) point mutations in the paravoltage-gated sodium channel (VGSC) gene [20], is well documented and largely spread across the country [11,21,22,23]. The mutation L1014F is largely predominant and has been reported to be involved in most cases of resistance to both DDT and type I pyrethroids [11,24,25,26]. The L1014S allele frequency is still low across the country and is considered to have a minor role in vector resistance [11]. The insensitive acetylcholinesterase (iAChe) G119S mutation in the *ace-1* gene [27] conferring resistance to both carbamates and organophosphates has currently not been reported across the country [28]. Several mutations such as the N1575Y [29] in the VGSC gene and different haplotypes have been reported [25,30]. Concerning metabolic resistance, several key detoxification genes were reported to be involved in mosquito resistance. These include the cytochrome P450-dependent monooxygenases (P450) *Cyp6p3*, *Cyp6m2*, *Cyp6p4*, and the glutathione-S-transferases *Gst1-6* [31]. Overexpression of additional genes implicated in insecticide resistance such as the oxidative decarboxylase *Cyp4G16* which catalyses epicuticular hydrocarbon biosynthesis, the superoxide dismutases *Sod3*, *Sod2*, *Gsts1-2*,and P450s *Cyp4h24*, *Cyp6P3*, *Cyp325c2* has also been recorded in pyrethroid-resistant *An. arabiensis* populations from North Cameroon [32]. Overexpression of P450s detoxification genes could be involved in resistance to bendiocarb (carbamate), pyrethroids, and a large number of compounds [28,31]. It is not known whether additional mechanisms such as cuticular modifications that delay insecticide uptake [33] are involved in insecticide resistance in *An. Gambiae* s.l. populations from Cameroon.

The present study focuses on the analysis of the insecticide resistance status, underlying molecular mechanisms conferring resistance in forest and city *An. gambiae* populations from Cameroon. This study also intends to examine cuticle alterations as an additional mechanism of insecticide resistance in Cameroon *An. gambiae* s.l. populations. 

## 2. Methods

### 2.1. Study Sites

Mosquitoes were collected in four localities in Cameroon: Sangmelima, Nyabessan, Mbandjock, and Yaoundé. All these localities are situated within the Congo-Guinean equatorial climate domain characterized by four seasons: Two rainy seasons (from March to June and from September to November) and two dry seasons (from December to February and from July to August). Yaoundé (3°52′ N; 11°31′ E) is a city of about 3 million inhabitants, occupying a surface area of over 400 km^2^. Although Yaoundé belongs to the equatorial forest domain, the landscape of the city is highly degraded with the extension of the city limits. The landscape of the city is characterized by an alternation of high and lowland areas frequently exploited for agriculture. Mosquito sampling in Yaoundé was conducted in the district of Bastos in the city center, a residential area with no practice of agriculture, and Nkolondom, a periurban district with large surface areas exploited for agriculture. Sangmelima (2°56′ N, 11°58′ E) is a small city of about 70,000 inhabitants located about 150 km south of Yaoundé in the heart of the equatorial forest with a high forest cover. Mbandjock (4°14′ N; 11°54′ E) is situated about 100 km north of Yaoundé in a semi-degraded forest domain at the limit with humid savanna. The city is situated close to the Sanaga River, the most important river in the country. Nyabessan (2°24′ N; 10°24′ E) is situated in the heart of the equatorial forest region close to the border with Equatorial Guinea. Nyabessan is situated close to the newly constructed Memve’ele dam and lies along the edges of the river Ntem. 

### 2.2. Collection of Mosquito Larvae, Rearing, and Processing

Field sampling of anopheline larvae and processing was conducted from September to November 2018 during the long rainy season in the five sites (Sangmelima, Nyabessan, Mbandjock, Yaoundé (Nkolondom, and Bastos)). Larval collections were undertaken in different habitats including temporary water collections, puddles, and semi-permanent sites to avoid over sampling single mosquito families. Collected samples from these different breeding sites were pooled per collection site and reared together in the insectary of the Malaria Research Laboratory of OCEAC. Larvae were fed with Tetramin^®^ fish food until pupae. Pupae were collected in cardboard cup and placed in netting cages for adult emergence. 

After emergence, adults were offered sugar solution until processing. A subset of 30–40 unexposed, non-blood fed, 3–5 days post eclosion/emergence female *An. gambiae* s.l. from different populations were preserved in RNA later for characterization of molecular mechanisms of insecticide resistance; that is knockdown resistance mutations and metabolic resistance genes. The same sample was also used for species identification. Another subset of about 120–150 mosquitoes per population were used for insecticide bioassays; survivors after exposure to insecticide were preserved in 70% alcohol and used for confirmation of molecular species identification.

### 2.3. Insecticide Bioassay 

Adult female *An. gambiae* s.l. reared from larval collections in different collection sites were tested alongside the susceptible laboratory strains Kisumu and Ngousso against three insecticides (deltamethrin, permethrin, and bendiocarb) following WHO guidelines [34]. 

*An. gambiae* s.l. females aged 3–4 days reared from larvae collected on the field were placed in batches of 20 to 25 mosquitoes per tube. The mosquitoes were then transferred to tubes with insecticide-impregnated papers and exposed for 1 h. The insecticide susceptible strains *An. gambiae* Kisumu and Ngousso were used as control to assess the quality of the impregnated papers. The number of mosquitoes knocked down by the insecticide was recorded after 1h of exposure; next, mosquitoes were fed with a 10% sugar solution and the number of dead mosquitoes was recorded 24 h post-exposure. Mosquitoes subjected to untreated papers were systematically included as controls. 

### 2.4. Mosquito Processing

#### 2.4.1. Total RNA and DNA Extraction from Mosquito Pools

Total RNA and DNA of a pool of female adult mosquitoes aged between 3–5 days were extracted using a magnetic beads-based approach with the MagSi kit (MagnaMedics Diagnostics B.V., Geleen, The Netherlands, Cat. No. MD01017). The quantity and purity of DNA and total RNA were assessed spectrophotometrically via Nanodrop measurements. The quality of RNA was assessed by 1.0% w/v agarose gel electrophoresis.

#### 2.4.2. Genotyping of Mosquito Samples and Multiplex RT-qPCR for Gene Expression Analysis

Species identification and target site mutation (L1014F, L1014S, N1575Y, G119S) determination were performed using the assays described in the IVCC Vector Population Monitoring Tool (VPMT) Protocol Manual [35] with modifications. Analysis of *An. gambiae* molecular forms was based on the insertion polymorphisms of SINE200 retrotransposons within speciation islands [36]. When analyzing mosquito pools, the % allele frequency for the previously mentioned traits was calculated with regression models using a methodology previously developed [37].

The previously described quantitative Reverse Transcription-real-time PCR (qRT-PCR) 3-plex TaqMan^®^ assays was used for the quantification of eight genes’ expression implicated in insecticide resistance (*Cyp6p3* [38], *Cyp6m2* [39], *Cyp9k1* [40], *Cyp6p4* [41], *Cyp6z1* [42], *Gste2* [43], *Cyp6p1* [44], and *Cyp4G16* [33]), using *RPS7* for normalization purposes in each assay [45].

A list of the primers and probes used in this study is provided in Appendix A. 

Reactions were performed in the Viia7 Real-Time PCR system (Applied Biosystems, Waltham, MA, USA) using a one-step RT-PCR master mix supplied by FTD (Fast-track diagnostics, Luxembourg) in a total reaction volume of 10 µL. The thermal cycle parameters were: 50 °C for 15 min, 95 °C for 3 min, and 40 cycles of 95 °C for 3 s and 60 °C for 30 s, allowing a sample to result time of ~75 min. Samples were amplified in at least two technical replicates, using eight biological replicates for allele frequency estimation and four biological replicates for gene expression analysis for each population. Each run always included a non-template control. The QuantStudio^TM^ Real-Time PCR system v1.3 (Applied Biosystems) software was used for the calculation of Ct values for each reaction, which were then used to calculate fold-changes according to the Pfaffl method [37] and allelic frequencies according to Mavridis et al. [37].

#### 2.4.3. Extraction and Quantification of Cuticle Hydrocarbon Lipids 

Female mosquitoes of susceptible control strains (Kisumu and Ngousso) and the field populations were collected and stored in an Eppendorf tube at −80°C until use. Before analysis, mosquitoes were air-dried at 25°C for 48 h and then pooled (20–25 female mosquitoes/replicate, 3 replicates for each population/strain), the dry weight of each replicate was measured, and Cuticular Hydrocarbon Lipids(CHC) analysis was performed by GC-MS and GC-FID as previously described in Balabanidou et al. [33] and Simma et al. [46]. Briefly, cuticular lipids from all samples were extracted by 1-min immersion in Hexane (x3) with gentle agitation; extracts were pooled and evaporated under a N2 stream. CHCs were separated from other components and concentrated prior to chromatography by Solid Phase Extraction (SPE). Quantitative amounts were estimated by co-injection of nC24 as an internal standard (2890 ng/mL in Hexane). CHC quantitation was calculated as the sum of area of 32 peaks in total, using the internal standard. 

### 2.5. Statistical Analysis

Calculation of 95% CIs and statistical significance was performed according to the Pfaffl method [3]. Graphs of metabolic gene expression were constructed with the SigmaPlot software (v12.0). The mortality rate was expressed as the ratio between the number of mosquitoes that were found dead or not capable to stand on their legs and exposed ones. Confidence intervals were computed using Medcalc. Comparison of mortality rate and fold-change were performed using chi square and Student *t*test, respectively.

## 3. Results

### 3.1. Bioassays Results

All bioassays were carried out using females of three to four days deriving from larval collection in different study sites (Sangmelima, Nyabessan, Mbandjock, Bastos, and Nkolodom) and two (2) susceptible *An. gambiae* laboratory strains (Kisumu and Ngousso) available in the insectary of OCEAC for more than 10 years. A total of 900 *An. gambiae* from the laboratory strain and 1050 from field populations were screened. 

The laboratory strains Kisumu and Ngousso were all susceptible to all compounds used with mortality ranging between 98–100%. The five collected field populations showed high resistance to pyrethroids. Mortality rate varied between 4–12% for deltamethrin 0.05% and 3–12% for permethrin 0.75%. For test with bendiocarb, the mortality rate varied from 66–86% (Table 1).

### 3.2. Species ID and Molecular Forms

A total of 280 specimens were genotyped to determine species present in each site. From the analysis, it appeared that in Bastos Yaoundé only *An. coluzzii* was recorded, whereas in Nkolondom *An. gambiae* was the only species present. In Nyabessan, Sangmelima, and Mbandjock, both *An. gambiae* and *An. coluzzii* were recorded (Table 2).

### 3.3. Screening of Target Site Mutations (kdr L1014F/S, kdr N1575Y, iAChe G119S) 

The distribution of different mutations associated with insecticide resistance was assessed. No mutations were detected in the susceptible laboratory strains. The L1014F allele was predominant in all sites with a frequency varying from 75.32% to 95.82% (Table 3 and Appendix A). The L1014S kdr allele was not detected in the populations from Bastos and Nkolondom in the city of Yaoundé, but was present in the remaining sites with a frequency varying from 1.38 to 23.05%. The N1575Y was recorded in Nyabessan, Nkolondom, and Mbandjock. The iAChe mutation G119S was recorded in almost all settings except in Bastos with a prevalence varying from 14.22 to 35.5% (Table 3 and Appendix A).

### 3.4. Expression Analysis of Genes Implicated in Insecticide Resistance

Quantitative PCR analyses were conducted to assess the expression profile of eight different genes involved in insecticide resistance. This includes *Cyp6p3*, *Cyp6m2*, *Cyp9k1*, *Cyp6p4*, *Cyp6z1*, *Gste2*, *Cyp6p1*, and *Cyp4G16*. High overexpression ratios were obtained when field populations were compared to the Kisumu strain (*An. gambiae*) originating from East Africa (Table 4 and Appendix A). On the other side, moderate overexpression ratios were obtained when field populations of either *An. gambiae* or *An. coluzzii* were compared to the Ngousso laboratory strain (*An. coluzzii*). The following genes *Cyp9k1*, *Cyp6m2*, *Cyp6z1*, and *Cyp4g16* always displayed the highest fold-changes in the different comparisons (Table 4 and Appendix A).

### 3.5. Analysis of Cuticular Hydrocarbon Lipids as a Marker of Pyrethroid Resistance

Analysis of CHCs and total mosquito HCs showed no significant quantitative increases in the CHC profiles in any of the field caught populations, compared to the susceptible control specimens (Appendix A). 

## 4. Discussion

The study aim was to characterize the profile and mechanisms conferring resistance in *An. gambiae* populations from South Cameroon forest settings. Both *An. gambiae* and *An. coluzzii* were present in the different settings. This was consistent with previous studies [4]. However, although sharing similar habitats, *An. gambiae* and *An. coluzzii* have been reported to segregate across different gradients: Distance from the coastal line and altitude [47]. *An. coluzzii* was also found to be more tolerant to ammonia and salinity in Cameroon compared to *An. Gambiae* [47,48]. The study indicated a multi-resistance profile in *An*. *Gambiae* s.l. populations from the South Cameroon forest region. A close association was recorded between phenotypic expression of resistance (low mortality rate) and genetic mechanisms involved with mosquito resistance to insecticides. A widespread distribution of target site mutations in both *An. gambiae* and *An. Coluzzii* was evident in the study’s populations. More precisely, the following mutations were recorded: *kdr*1014F, *kdr* 1014S, N1575Y, iAChe (G119S).

For the first time the G119S mutation conferring resistance to both carbamates and organophosphates was recorded in Cameroon. This mutation is largely spread in West Africa and has been reported in different studies [49,50,51,52,53,54,55,56]. Whether the emergence of this mutation in Cameroon is a de novo phenomenon which appeared independently in forest *An. gambiae* populations or if this is the result of intense gene flow between *An. gambiae* populations warrants further investigations. The mutation was not recorded in *An. coluzzii* populations and this probably suggests that the mutation could have emerged recently in the country. The mutation is still absent from several countries in Central Africa [40,44,57], while in West Africa, *An. gambiae*, *An. coluzzii*, and *An. arabiensis* have been reported to carry the allele [53,58] with evidence of duplication of the Ace-1 gene to generate fixed heterozygotes [58,59]. It was also reported in *An. gambiae* that duplication produces extremely resistant and multiple resistant phenotypes [60]. In *Culex* sp. it was demonstrated that duplication reduces the high fitness cost associated to G119S mutation [61]. 

The presence of different resistance mechanisms in mosquito populations from forested zones (Appendix A), support intense selective pressure taking place on the field. In addition to treated nets, the use of pesticides in agriculture could constitute the predominating factors selecting for resistance [17,19,62]. During the last decade intense deforestation has given way to the practice of agriculture with different crops being cultivated year-round. The rapid deforestation has provided significant opportunities for the expansion of the malaria vector *An. gambiae* which is now highly distributed in the area [4,47]. The cultivation of crops such as tomatoes, cabbages, vegetables, and new cultural practices in wetland requiring huge amounts of pesticides have set ground for an increasing selection of vector populations [16,19]. Agricultural practices create numerous trenches that retain water from rain and irrigation systems. *An. gambiae* s.l. females were reported to often lay their eggs in breeding sites located around agricultural settings suggesting that larvae may undergo a selection pressure from agricultural pesticides that promotes the emergence of resistance [24]. Studies conducted in agricultural cultivated sites indicated the frequent use by farmers of cocktail insecticides containing organochlorine, pyrethroids, carbamates, and organophosphates for fighting against pests [16,19,24]. In West Africa, studies conducted in agricultural cultivated sites indicated a high usage rate of pesticides and a high level of resistance in vector populations and the prevalence of different resistant mechanisms [60,62,63].

The *kdr*West allele (1014F) was found in very high frequencies across the different studied populations, whereas *kdr*East mutation (1014S) was found in low frequencies. These findings were in accordance with previous reports suggesting the high distribution of the L1014F allele across the country [11]. The kdr mutation was reported to be closely associated with resistance to DDT and permethrin in Cameroon [24]. The N1575Y mutation on the other side was recorded at a low frequency in the different sites. This mutation has been previously reported to increase the level of resistance of *An. gambiae* populations to type 2 pyrethroids in West Africa [29]. This mutation as well as the L1014S mutation were less frequent in *An. coluzzii*.

Eight insecticide resistance related genes (seven detoxification genes and one oxidative decarboxylase that catalyzes epicuticular hydrocarbon biosynthesis) among those reported in previous studies in Cameroon and West Africa, were characterized in the present study. Five of the eight genes screened including *Cyp6m2*, *Cyp9k1*, *Cyp6p4*, *Cyp6z1*, and *Cyp4g16* were overexpressed in at least one population. *Cyp9k1* showed the greatest folds of upregulation (up to 8.74-folds) and was found to be upregulated in most (4/5) of the study’s populations. *Cyp9k1* was recently reported in Equatorial Guinea to be involved with increased metabolism of deltamethrin [40]. Functional characterization of this gene indicated that it could metabolize both deltamethrin and pyriproxifen but this gene was not found to metabolize bendiocarb [40]. This gene was recorded highly overexpressed in DDT and pyrethroid-resistant *An. gambiae* populations from Cameroon [31]. *Cyp6m2* was also found to be intensively upregulated (up to 5.27 folds) in 3/5 populations. This gene has been recorded in previous studies in DDT-resistant *An. gambiae* populations [31]. *Cyp6m2* alongside *Cyp9k1*, *Cyp6p4* have been previously found to be upregulated in DDT and pyrethroid-resistant *An. gambiae* mosquitoes from Cameroon [31]. *Cyp6m2* expression in Drosophila was found to generate Drosophila phenotypes-resistant to bendiocarb, DDT, and class I and II pyrethroids [60]. This gene was rather not found to be associated to resistance to bendiocarb in Cameroon [28]. *Cyp6p3* was also recorded highly overexpressed in most populations, *Cyp6p3* overexpression has always been linked with pyrethroid resistance in *An. gambiae*. Recombinant *Cyp6p3* enzymes was found to metabolize bendiocarb in vitro [60]. *Cyp6z1* and *Cyp6p4* were additionally found to be upregulated also in bendiocarb-resistant *An. gambiae* mosquitoes from Cameroon [5]. *Cyp4g16* recorded overexpressed, is a functional oxidative decarboxylase gene known to catalyze epicuticular lipid biosynthesis [33]. It is also considered to contribute to insecticide resistance via the enrichment of the CHC content, thus reducing pyrethroid uptake [33].

In contrast to previous studies, which showed that the epicuticle CHC were enriched in pyrethroid resistance *An. Gambiae* [33] and *An. arabiensis* [46] female mosquitoes, we did not detect any significant increase of CHC in any of the populations analyzed, compared to the susceptible strains analyzed, possibly indicating that this type of cuticle resistance (epicuticular thickening due to elevated CHC) may not play a major role in the phenotype.

## 5. Conclusions

The study suggested multiple mechanisms involved in pyrethroid and carbamate-resistant *An. gambiae* populations from the South Cameroon forest region. If no action is taken, the rapid escalation of insecticide resistance could ultimately result into control failure. There is therefore an urgent need for more studies to assess the operational impact of insecticide resistance on LLINs efficacy in different epidemiological settings in order to preserve this tool. Cameroon launched a third nationwide campaign of free distribution of LLINs during the month of June 2019. This campaign as the previous ones still saw the distribution of first generation LLINs to the population despite evidences of their poor efficiency against pyrethroid-resistant mosquitoes. It becomes urgent that strategies to mitigate insecticide resistance, such as the replacement of current treated nets by new generation nets treated with alternative insecticides or a mixture of insecticides, be implemented for a better management of insecticide resistance. Moreover, integrated control strategies need to be implemented on the field for malaria control and elimination in Cameroon. 

## Figures and Tables

**Table 1 genes-10-00741-t001:** Mortality rate of field collected and laboratory strains *Anophelesgambiae* s.l. exposed to pyrethroid (deltamethrin 0.05% and permethrin 0.75%) and carbamate (bendiocarb 0.01%) insecticides.

			Pyrethroid		Carbamate
Site Characteristics	Populations	N	Deltamethrin	Permethrin	Bendiocarb
**Susceptible strain**	Kisumu lab strain	100	100	100	100
	Ngoussolab strain	100	100	98	100
**Urban site without agricultural activities**	Bastos	100	5.00	3.00	86.00
**Urban site with agricultural activities**	Nkolondom	100	8.00		66.00
**Forest**	Nyabessan	100	4.00	4.00	74.00
	Sangmelima	100	5.00	6.00	82.00
	Mbandjock	100	11.2	6.0	66.00

**Table 2 genes-10-00741-t002:** Distribution of *An. Gambiae* s.l. species in the different study sites.

Populations/Strains	Species	%
Kisumu laboratory strain	*An. gambiae*	100%
Ngousso laboratory strain	*An. coluzzii*	100%
Bastos (Yaoundé)	*An. coluzzii*	100%
Nkolondom (Yaoundé)	*An. gambiae*	100%
Sangmelima	*An. gambiae*	59.98%
	*An. coluzzii*	40.02%
Mbandjock	*An. gambiae*	65.36%
	*An. coluzzii*	34.66%
Nyabessan	*An. gambiae*	90.56%
	*An. coluzzii*	9.44%

**Table 3 genes-10-00741-t003:** Incidence of resistance alleles in different populations of *An. Gambiae* s.l. mosquitoes.

Populations	Sample Size (Alleles)	Resistant Mutation Allelic Frequencies (Mean ± SE)			
		Pyrethroids			Carbamates/Organophosphates
		% kdr L1014F	% kdr L1014S	% kdr N1575Y	% iAChe G119S
Kisumu lab strain	80	0	0	0	0
Ngusso lab strain	80	0	0	0	0
Bastos	80	81.33 ± 5.1	0	0	0
Nyabessan	80	75.32 ± 1.6	23.05 ± 2.07	5.28 ± 3.1	14.22 ± 8.85
Sangmelima	80	90.06 ± 4.62	1.38 ± 1.38	0	20.25 ± 7.06
Nkolodom	80	95.82 ± 1.53	0	9.46 ± 3.2	21.78 ± 12.8
Mbandjock	80	89.06 ± 4.05	1.71 ± 1.71	12.87 ± 3.2	35.5 ± 4.3

**Table 4 genes-10-00741-t004:** Gene expression analysis in the five resistant mosquito populations compared to the susceptible strains. Bold letters indicate statistically significant upregulation, asterisks (*) indicate consistent upregulation compared to both susceptible strains.

Population		Fold Changes (95% CI)							
		*CYP6P3*	*CYP6M2*	*CYP9K1*	*CYP6P4*	*CYP6Z1*	*GSTE2*	*CYP6P1*	*CYP4G16*
Bastos	vs KIS	4.21	4.39 *	7.30 *	3.74 *	3.36 *	2.46	0.85	1.71
		(2.63–6.96)	(2.93–6.10)	(4.03–19.5)	(2.56–5.09)	(2.36–4.93)	(1.99–3.12)	(0.71–0.97)	(1.20–3.76)
	vsNG	1.18	2.70 *	2.68 *	2.95 *	2.42 *	0.71	0.45	1.18
		(0.80–1.78)	(1.73–3.75)	(1.76–5.71)	(1.93–4.17)	(1.64–3.42)	(0.56–0.87)	(0.36–0.56)	(0.92–1.55)
Nyabessan	vs KIS	0.94	1.07	2.55	0.49	2.01	0.74	0.90	2.04
		(0.34–1.97)	(0.31–2.33)	(1.15–8.50)	(0.31–0.69)	(1.28–3.33)	(0.47–1.21)	(0.75–1.15)	(1.07–5.33)
	vsNG	0.27	0.66	0.94	0.39	1.45	0.21	0.48	1.41
		(0.10–0.51)	(0.20–1.44)	(0.49–2.49)	(0.24–0.57)	(0.89–2.31)	(0.13–0.34)	(0.38–0.64)	(0.86–2.27)
Sangmelima	vs KIS	0.63	1.64	4.15 *	1.26	1.79	1.37	1.24	3.11 *
		(0.24–1.24)	(0.85–4.40)	(2.22–10.7)	(0.61–3.03)	(1.14–2.92)	(1.07–1.80)	(1.01–1.42)	(2.31–6.49)
	vsNG	0.18	1.01	1.52 *	0.99	1.29	0.40	0.66	2.15 *
		(0.07–0.32)	(0.50–2.71)	(1.03–2.26)	(0.46–2.48)	(0.80–2.03)	(0.30–0.50)	(0.51–0.82)	(1.76–2.66)
Nkolodom	vs KIS	1.71	5.27 *	8.74 *	3.17 *	2.79	1.56	0.835	2.65 *
		(0.81–3.68)	(1.72–14.9)	(3.07–28.4)	(1.18–5.93)	(0.99–6.01)	(1.34–2.06)	(0.71–1.10)	(1.70–6.83)
	vs NG	0.48	3.24 *	3.21 *	2.50 *	2.01	0.451	0.411	1.83 *
		(0.25–0.96)	(1.08–9.18)	(1.29–8.33)	(1.09–5.76)	(0.735–4.17)	(0.38–0.60)	(0.36–0.62)	(1.30–2.61)
Mbandjock	vs KIS	1.51	3.31 *	6.26 *	1.89	2.13 *	0.98	1.01	1.85
		(0.73–2.75)	(2.07–4.59)	(3.25–16.1)	(1.11–2.78)	(1.39–3.30)	(0.69–1.26)	(0.77–1.16)	(1.301–4.08)
	vs NG	0.42	2.04 *	2.30 *	1.49	1.53 *	0.29	0.53	1.28
		(0.22–0.71)	(1.22–2.92)	(1.37–4.71)	(0.84–2.32)	(1.01–2.32)	(0.19–0.35)	(0.39–0.67)	(0.99–1.67)

KIS: Kisumu susceptible laboratory strain; NG: Ngousso susceptible laboratory strain. Statistical significance was assessed according to the Pfaffl method [37].

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
