# Peer review of "Status of Insecticide Resistance and Its Mechanisms in Anopheles gambiae and Anopheles coluzzii Populations from Forest Settings in South Cameroon"

_genes, 2019, doi:10.3390/genes10100741_

Round 1

Reviewer 1 Report

The manuscript entitled “Status of insecticide resistance and its mechanisms in An. gambiae from forest settings in South Cameroon” is well written with clear objective.  The aim of this study was to characterize the profile and mechanisms conferring resistance in An. gambiae populations from different parts of Cameroon and the authors succeeded to achieve it using the appropriate methodology and analyzing the results properly. The actual findings of this study are also very useful and should be taken into account when the control strategies for malaria control and its eventual elimination will be designed.

Author Response

Thank you very much for your comments.

Reviewer 2 Report

Bamou et al. presented an important study to investigate the insecticide resistance status and mechanisms in An. gambiae populations under forest settings in South Cameroon, which is one of the 11 countries most affected by malaria in the world. Authors used a comprehensive strategy combining bioassay, SNP detection and differential expression with molecular markers, and quantifying cuticle hydrocarbon lipids to evaluate resistance levels of An. gambiae to pyrethroids and carbamate as well as the mechanisms underlying resistance. Basically, it is a scientific sound study and will be interested to readers in biology, molecular biology, toxicology, and vector control. I do have some comments for authors improving their MS in general.

Introduction

Authors should provide more information to describe the background of pyrethroid and carbamate insecticides usage for vector control in the world and Cameroon.

Should cite more papers to provide sufficient background for the markers used in the diagnostic study. For example, cite the functional study papers for these kdr mutations, L1014F, L1014S, N1575Y and mutation G119S (Edi et al. 2014 PloS Gene) instead only cite papers that used these mutations as markers for diagnostic studies. Similarly, cite the functional/original studies for these detoxification genes, such as CYP6P3 (Muller et al. 2008 PloS Genet), CYP6M2 (Mitchell et al. 2012 PNAS)…some review papers can be helpful too (Liu 2015 Annu Rev Ent; Zhu et al. 2016 Insects)

CYP4G16 is not involved in detoxification but is a functional oxidative decarboxylase (Balabanidou et al. 2016 PNAS)

Materials and Methods Mosquito processing Total RNA and DNA extraction from mosquito pools

Details for the age, sex, developmental stage of these mosquito samples that were used for RNA extraction are needed. Because this information is critical for the gene expression of the molecular markers used for this study (Table 4, Fig. S2).

2.4.2 Genotyping of mosquito samples and multiplex RT-PCR for gene expression analysis

More details for SNP detection and quantitative RT-PCR are required here. For example, primer information used for TaqMan assays? How many replicates (biological and technique replicates)? How to calculate the expression levels for these detoxification genes? What’s the PCR program used for SNP detection? Any difference between two mosquito species?

Results Table 2. Bastos population has only An coluzzii (100%)? If it is true, why it was still used for other studies (Table 3 &4) to evaluate An. gambiae resistance and mechanisms? If look at the Table legends of 3&4 and the title of this MS, only An. gambiae was studied. But actually both An gambie and An coluzzii were studied. 4 Screening of detoxification gene expression

CYP4G16 is not a detoxification gene. Should mention which genes are associated with pyrethroid resistance and which genes are related to carbamate or OP resistance

Table 4, asterisks stand for P value <0.05? student t-test? Please clarify it. 5 Analysis of cuticular hydrocarbon lipids, as a marker of pyrethroids resistance:

Does the differential expression of CYP4G16 serve as a marker too? Or the thickness of the cuticle?

Adding a summary table or figure to sum up all possible mechanisms in each population will be helpful

Discussion

This is the section needed to be improved significantly.

Page 7, lines 230-231, “For the first time the G119S mutation conferring resistance to both carbamates and organophosphates was recorded”- what does this sentence mean? Does the conclusion come from this study or from previous studies? Please clarify it. Page 7, lines 235-236, “The mutation was not recorded in An. coluzii populations and probably suggests a recent emergence of the mutation in the country”. This statement does not make any sense. Page 8, line 242, this multi-resistance pattern is not clear. A summary table or figure of all results will be helpful for this statement. Page 8, lines 242-252, authors can discuss a little more of the contribution of agricultural activities to pesticide resistance of disease vectors, using the result from Table 1 Page 8, line 260, these eight P450/GST genes are not all detoxification genes, CYP4G16 is a functional oxidative decarboxylase, need more discussion for this point Discuss the link of phenotype (Table 1) and genotype (Tables 3&4). Besides target-site insensitivity, enhanced detoxification, any other possible mechanisms involved?

Author Response

Reviewer ≠2

Introduction

Query 1: Authors should provide more information to describe the background of pyrethroid and carbamate insecticides usage for vector control in the world and Cameroon.

Response 1: Done

Query 2: Should cite more papers to provide sufficient background for the markers used in the diagnostic study. For example, cite the functional study papers for these kdr mutations, L1014F, L1014S, N1575Y and mutation G119S (Edi et al. 2014 PloS Gene) instead only cite papers that used these mutations as markers for diagnostic studies. Similarly, cite the functional/original studies for these detoxification genes, such as CYP6P3 (Muller et al. 2008 PloS Genet), CYP6M2 (Mitchell et al. 2012 PNAS)…some review papers can be helpful too (Liu 2015 Annu Rev Ent; Zhu et al. 2016 Insects)

Response 2: Done; we have included references citing functional studies for each target site and insecticide-resistance related genes.

Query 3: CYP4G16 is not involved in detoxification but is a functional oxidative decarboxylase (Balabanidou et al. 2016 PNAS)

Response 3:  We thank the reviewer for this comment and giving us the opportunity to correct this information throughout the manuscript.

Materials and Methods Mosquito processing Total RNA and DNA extraction from mosquito pools

Query 4: Details for the age, sex, developmental stage of these mosquito samples that were used for RNA extraction are needed. Because this information is critical for the gene expression of the molecular markers used for this study (Table 4, Fig. S2).

Response 4: Done. We have added the following part in the manuscript:

“A subset of 30-40 unexposed, non-blood fed, 3-5 days post eclosion/emergence females An gambiae sl from different populations were preserved in RNAlater for characterisation of molecular mechanism of insecticide resistance, that is knockdown resistance mutations and metabolic resistance genes. 2.4.2 Genotyping of mosquito samples and multiplex RT-PCR for gene expression analysis”

Query 5: More details for SNP detection and quantitative RT-PCR are required here. For example, primer information used for TaqMan assays? How many replicates (biological and technique replicates)? How to calculate the expression levels for these detoxification genes? What’s the PCR program used for SNP detection? Any difference between two mosquito species?

Response 5: Done, we have clarified this part in our revised manuscript:

“The thermal cycle parameters were: 50 °C for 15 min, 95 °C for 3 min, and 40 cycles of 95 °C for 3 sec and 60 °C for 30 sec, allowing a sample to result time of ~75 min. Samples were amplified in at least two technical replicates, using eight biological replicates for allele frequency estimation and four biological replicates for gene expression analysis for each population. Each run always included a non-template control. The QuantStudioTM Real-Time PCR system v1.3 (Applied Biosystems) software was used for the calculation of Ct values for each reaction, which were then used to calculate fold-changes according to the Pfaffl method and allelic frequencies according to Mavridis et al.”

We have also provided the list of primers and probes used in the study as supplementary material.

Query 6: Results Table 2. Bastos population has only An coluzzii (100%)? If it is true, why it was still used for other studies (Table 3 &4) to evaluate An. gambiae resistance and mechanisms? If look at the Table legends of 3&4 and the title of this MS, only An. gambiae was studied. But actually both An gambie and An coluzzii were studied.

Response 6:  The reviewer is correct. Both An. gambiae (An. gambiae s.s. form S) and An. coluzzii (An. gambiae s.s. form M) were present in the different settings, both belonging to the An. gambiae s.l. species complex. Following the reviewer’s suggestion, we have added An. coluzzii in the title and in the Tables indicated to denote that.

4 Screening of detoxification gene expression

Query 7: CYP4G16 is not a detoxification gene. Should mention which genes are associated with pyrethroid resistance and which genes are related to carbamate or OP resistance

Response 7:  We have corrected the information regarding the role of CYP4G16 throughout our revised manuscript (please see previous comment). Regarding the role of the remaining genes, most of them are implicated in pyrethroid/DDT resistance, while some of them have been additionally, associated with resistance to bendiocarb (Cyp6p4, Cyp6z1, Cyp6p3). We included this information in our Discussion.

Query 8: Table 4, asterisks stand for P value <0.05? student t-test? Please clarify it.

Response 8: Statistical significance was assessed according to the Pfaffl method. We have added this information in Table 4.

5 Analysis of cuticular hydrocarbon lipids, as a marker of pyrethroids resistance:

Query 9: Does the differential expression of CYP4G16 serve as a marker too? Or the thickness of the cuticle?

Response 9: We thank the reviewer for this comment. Quantitative modification of cuticular hydrocarbons is associated with increased expression of Cyp4g16 and could potentially serve as a marker. However, in our study we detected Cyp4g16 overexpression in some populations but we did not detect any significant increase of CHC in any of the populations analyzed, possibly indicating that this type of cuticle resistance (epicuticular thickening, due to elevated CHC) may not play a major role in the phenotype.

Query 10: Adding a summary table or figure to sum up all possible mechanisms in each population will be helpful

Response 10: We thank the reviewer for this suggestion. In our revised manuscript, we have included the following table that summarizes this information:

Table S3 Summary of all resistance mechanisms for each of the study’s populations

Population

Bastos (Yaoundé)

Nkolondom (Yaoundé)

Sangmelima

Mbandjock

Nyabessan

Resistance Mechanism

Target site mutations

kdr L1014F/S

kdr L1014S

kdr N1575Y

iAChE G119S

Metabolic and CHC biosynthesis gene overexpression

CYP6P3

CYP6M2

CYP9K1

CYP6P4

CYP6Z1

GSTE2

CYP6P1

CYP4G16

Discussion

This is the section needed to be improved significantly.

Query 10: Page 7, lines 230-231, “For the first time the G119S mutation conferring resistance to both carbamates and organophosphates was recorded”- what does this sentence mean? Does the conclusion come from this study or from previous studies? Please clarify it.

Response 10: This mean that compared to previous study where the same mutation were screened (Antonio-Nkondjio et al 2016 Malar J; Antonio-Nkondjio et al 2017 Parasites &Vectors), this study is the first and unique showing it presence in Anopheles gambiae sl population in the country (Cameroon).

The sentence was modified as follow “For the first time the G119S mutation conferring resistance to both carbamates and organophosphates was recorded in Cameroon”

Query 11: Page 7, lines 235-236, “The mutation was not recorded in An. coluzii populations and probably suggests a recent emergence of the mutation in the country”. This statement does not make any sense.

Response 11: The sentence was changed to “The mutation was not recorded in An. coluzzii populations and this probably suggests that the mutation could have emerged recently in the country.”

Query 12: Page 8, line 242, this multi-resistance pattern is not clear. A summary table or figure of all results will be helpful for this statement.

Response 12: A summary table was inserted in the draft showing all resistance pattern (Table S2). Please, see comment of query 10

Query 13: Page 8, lines 242-252, authors can discuss a little more of the contribution of agricultural activities to pesticide resistance of disease vectors, using the result from Table 1

Response 13: Done

Query 14: Page 8, line 260, these eight P450/GST genes are not all detoxification genes, CYP4G16 is a functional oxidative decarboxylase, need more discussion for this point Discuss the link of phenotype (Table 1) and genotype (Tables 3&4). Besides target-site insensitivity, enhanced detoxification, any other possible mechanisms involved?

Response 14: The function of all these eight P450/GST genes was clarified in the revised version. Seven were detoxification genes and one (CYP4G16) oxidative decarboxylase that catalyzes epicuticular hydrocarbon biosynthesis.

Reviewer 3 Report

Comments to the manuscript genes-575546 intended as paper in Genes entitled “Status of insecticide resistance and its mechanisms in An. gambiae from forest settings in South Cameroon” by Roland Bamou, Nadège Sonhafouo-Chiana, Konstantinos Mavridis, Timoléon Tchuinkam, Charles Wondji, John Vontas and Christophe Antonio-Nkondjio which elucidate the molecular background of insecticide resistance in malari mosqutoes in Africa.

This paper is an original, comprehensive, well-structured, solid contribution to the study of insecticide resistance focusing on mutation in important resistance genes as well as expression of detoxification genes, but the authors need to make a few points more clear to the reader.

The abstract should be made concise and general informative. There are too many details, like sampling sites, range of frequencies etc.

The Introduction on the other hand are missing detailed information.

Ln70-78; you mention multiple mutations, but not the genes! Additionally the mutations from the various genes are mixed when mentioned.

Ln78-83; the function the various genes are an enigma. Please add information.

Ln178 Results; do you know anything about the differences in toxicology between different species of Anopheles? Some of the populations in Table 1 are one species other populations are a mix, does that affect the results? What is the unit in Table 1?

Ln189; you make species identification, but we are missing the background for doing this. Please delete from Results or add the background to the Introduction about An. gambia, An coluzzi, An arabiensis….

Ln195, Table 3: what is iAche compared to Ache? Are there difference in frequency of different resistance mutations in the two species identified?

Table 4: The detox data would be much better presented as two aligned column figures. Please make column figure with error bars.

Discussion: a more comprehensive effort should be given to the various Anopheles species. What are common and what are different in relation to resistance genes.

Author Response

Reviewer ≠3

Query 1: The abstract should be made concise and general informative. There are too many details, like sampling sites, range of frequencies etc.

Response 1: Done. Details were removed from the abstract accordingly

The Introduction on the other hand are missing detailed information.

Query 2: Ln70-78; you mention multiple mutations, but not the genes! Additionally the mutations from the various genes are mixed when mentioned.

Response 2:  Following the reviewer’s indication we now mention the genes for each mutation in our revised introduction (kdr L1014F, L1014S, N1575Y: point mutations in the para voltage-gated sodium channel, iAChe : insensitive acetylcholinesterase G119S point mutation in the ace-1 gene).

Query 3: Ln78-83; the function the various genes are an enigma. Please add information.

Response 3: Following the reviewer’s suggestion, we now mention the function of these genes in the revised manuscript:

Concerning metabolic resistance, several key detoxification genes were reported to be involved in mosquito resistance by metabolizing organochlorine and pyrethroids in the city of Yaoundé, include the cytochrome P450-dependent monooxygenases (P450) Cyp6p3, Cyp6m2, Cyp6p4, and the glutathione-S-transferases Gst1-6. Overexpression of additional genes implicated in insecticide resistance such as the oxidative decarboxylase Cyp4G16 which catalyses epicuticular hydrocarbon biosynthesis, the superoxide dismutases Sod3, Sod2, Gsts1-2, and P450s Cyp4h24, Cyp6P3, Cyp325c2 has also been recorded in pyrethroid resistant An. arabiensis populations from north Cameroon.

Query 4: Ln178 Results; do you know anything about the differences in toxicology between different species of Anopheles? Some of the populations in Table 1 are one species other populations are a mix, does that affect the results? What is the unit in Table 1?

Response 4: Table 1 is showing the mortality rate (in percentage) of each mosquito’s population (Anopheles gambiae sl) tested with different insecticides (Permethrin 0.75%, Deltamethrin 0.05% and bendiocarb 1%). We added in the introduction a sentence providing information on factors influencing the distribution of the two species. We have no idea about the difference in toxicology between species of Anopheles gambiae complex (An. coluzzi and An. gambiae ss) and we think that it does not affect results of the Table1.

Query 5: Ln189; you make species identification, but we are missing the background for doing this. Please delete from Results or add the background to the Introduction about An. gambia, An coluzzi, An arabiensis….

Response 5: Done. This sentence was added in introduction “In Cameroon members of the An. gambiae complex consist of An. gambiae, An. coluzzii, An. melas and An. arabiensis but only An. gambiae and An. coluzzii have been reported from forest settings [57, 68].”

Query 6: Ln195, Table 3: what is iAche compared to Ache? Are there difference in frequency of different resistance mutations in the two species identified?

Response 6: We thank the reviewer for giving us the opportunity to clarify this issue. Ache, iAChe and Ace-1R refer to the same mutation (G119S) of the Ace-1 gene that confers resistance to carbamates and OPs. Following the reviewer’s’ comment, we have used iAChe to denote the G119S mutation consistently in our revised manuscript. Protocol used in this study is unable to provide frequency of each species but the absence of the G119S in Bastos suggest the difference in the frequency in the two species for this trait/gene. Species identity may have no or little influence in other mutation frequencies observed in this study.

Query 7: Table 4: The detox data would be much better presented as two aligned column figures. Please make column figure with error bars.

Response 7: Not done because this figure is provided in supplementary material as Figure S2.

Query 8: Discussion: a more comprehensive effort should be given to the various Anopheles species. What are common and what are different in relation to resistance genes.

Response 8: Done

All requested changes were done accordingly and are highlighted in the main text.

Round 2

Reviewer 2 Report

Authors answered most of my questions properly. I recommend publishing this study in Genes.

One more comment:

Page 14, line 306, The TaqMan assay you used for mutation detection is a very sensitive method to identify mutations even when the frequency is still very low. Therefore, the absent of G119S mutation may suggest target-site mutation is not the major mechanisms contributing OP/Carbamate resistance in An. coluzzii populations of Cameroon.